# The role of perceived expertise and trustworthiness in research study and clinical trial recruitment: Perspectives of clinical research coordinators and African American and Black Caribbean patients

Susan E. Morgan[1]*, Tyler R. Harrison[1], Kallia O. Wright[1], Xiaofeng Jia[2], Bonnie Deal[2], Kate Malova[3]

1 Communication Studies Department, School of Communication, University of Miami, Coral Gables, FL, United States of America, 2 School of Communication, University of Miami, Coral Gables, FL, United States of America, 3 Simon Business School, University of Rochester, Rochester, NY, United States of America

* semorgan@miami.edu

**Data Availability Statement:** The anonymized dataset used for this study can be found in the

## Abstract

This study investigates the role of source credibility on minority participant recruitment, particularly African American and Black Caribbean patients. A total of nine focus groups (N = 48 participants) were conducted with both patient groups and clinical research coordinators (CRCs). Using the elaboration likelihood model as a guiding framework for analysis, this study found that the credibility of research coordinators (or other professionals who recruit for research studies and clinical trials) was instrumental in shaping attitudes of prospective participants. The perspectives of patients and CRCs aligned closely, with few exceptions. For both groups, professionalism and professional displays (clothing, institutional artifacts) enhanced perceived expertise, a core component of credibility. Trustworthiness, another important component of credibility, was fostered through homophily between recruiter and patient, expressions of goodwill and assuaging anxiety about CRCs' financial motivations for recruitment. Additionally, CRCs believed that credibility was supported when CRCs could emphasize transparency and truthfulness in communication. The importance of these findings for the development of empirically-based training programs to improve communication practices in recruitment contexts is discussed.

## Introduction

Medical innovations that offer safe and effective treatments for illness cannot reach the public without thorough evaluation through clinical trials. Unfortunately, patients are rarely offered the opportunity to join a clinical trial; a nationally representative sample of diverse adults in the U.S. indicates that just 9% have ever been presented with the opportunity to join a clinical trial [1–3]. While African Americans are approached more frequently to join trials (16%), they are also less likely to accept those invitations [2]. The reasons for this disproportionate lack of

repository at the Library of the University of Miami: https://doi.org/10.17604/m4g4-sm36.

**Funding:** This study received funding support from the University of Miami Clinical Translational Science Institute, UL1 TR000460 (SEM), http://miamictsi.org, and the University of Miami's Provost's Research Award, PRA2022-2510 (SEM, TRH), https://www.research.miami.edu/about/admin-areas/rde/provosts-awards/provosts-research-awards/index.html. The funders had no role in study design, data collection and analysis, decision to publish, or preparation of the manuscript.

**Competing interests:** The authors have declared that no competing interests exist.

representation in research studies include lack of medical insurance, mistrust of the health care system, and lack of culturally appropriate communication behaviors during recruitment [4, 5]. Lower rates of Black/African American representation contribute to health disparities for two primary reasons: (1) clinical trials, particularly Phase I trials, offer access to cutting-edge care and treatments that cannot be accessed in any other way, particularly for those who are uninsured; (2) medical treatments of diseases that affect minority populations that have not been tested within those populations may not be as effective because of biological differences in responses to these treatments. While there is a growing body of work on factors influencing study participation among members of African American and other minoritized communities, there are no studies that attend to issues affecting research participation decisions by Black Caribbean patients. Including Black Caribbean populations in these studies is important because they represent a substantial sub-population in urban areas, including places like South Florida and northeastern cities like New York.

The empirical evidence for a set of best practices for clinical trial communication is still emerging. What has been established is mostly (but not entirely) atheoretical. The elaboration likelihood model (ELM) is among the best-known theories that examine the influence of credibility and persuasion [6], along with the competing heuristic systematic model (HSM) [7]. While there are nuanced differences in the way these two models propose the ways in which existing attitudes affect subsequent attitude formation and the role of motivations in processing information, we have elected to use ELM and its refinement, the Unimodel, as a framework for analyzing our data because of its more widespread applications, particularly in health contexts. Our primary goal is to identify the role of the credibility (particularly perceived expertise and trustworthiness) of clinical research coordinators and other medical staff in motivating patients to meaningfully consider joining a clinical trial or research study and to identify the communication factors that play a role in successful recruitment to clinical trials and research studies.

## The elaboration likelihood model and clinical trial recruitment

The elaboration likelihood model posits that receivers (e.g. patients who are being approached about participating in a research study or clinical trial) cognitively process messages in one of two ways: centrally or peripherally. Central route processing involves careful consideration of the content of the message itself, while peripheral route processing uses other types of information, that is not as deeply analyzed, to form attitudes about the proposed action (e.g., study participation), including characteristics of the source of the message (e.g., the study staff member presenting the study) [8].

Whether patients engage in central route or peripheral route processing depends largely on whether the patient is motivated to think about the information being presented [8]. Motivations to process information are higher when a receiver views the issue as personally important [9]; thus, individuals who are asked to join a study related to a rare and difficult-to-treat disease that they themselves have are likely to attend more carefully to the information about the study, while those who are asked to participate in research designed to prevent a disease that the patient is not at high risk for contracting (i.e., have lower levels of involvement) are less likely to be motivated to process study information through the central route. The type of processing can also be impacted by the source of the information and the source's credibility, the focus of this study. Interestingly, while characteristics of the source are often treated as a peripheral cue, research has shown that source credibility can nonetheless motivate central route processing [10].

In addition to issue involvement, the ability to process complex medical information plays an important role in whether or not patients rely on central or peripheral route processing.

Peripheral processing due to lower cognitive capacity is more likely, for instance, in patients who have a lower level of education (which correlates with lower levels of health literacy) [11], and those who are emotionally or cognitively overburdened, perhaps because they have to make a series of difficult decisions in the face of a frightening diagnosis.

Whether an individual engages in central versus peripheral route processing matters to the duration of attitude change, with central route processing being associated with stronger and more meaningful change [8]. A common critique of ELM is that the theory seems to imply that there is orthogonality in the categories of peripheral and central routes [12], but information processing can take place on a continuum between these two routes [8, 13]. Additionally, information processing can be quite dynamic; under conditions of distraction, individuals are more likely to use peripheral cues to form short-term attitudes and even make behavioral decisions. When those distractions are no longer an issue, more careful attention can be paid to important details, with a greater possibility of a more enduring attitude change emerging [8]. However, the ELM framework is nonetheless useful for the current study because it is both widely used and offers insights that are highly relevant for the identification of communication behaviors that can help motivate patients to thoughtfully consider the possibility of clinical trial participation. Moreover, findings using this framework can point to areas that could be the focus of professional training and development that can eventually lead to improvements in clinical trial accrual.

The literature on clinical trial communication factors specifically related to the characteristics of the person presenting information about research participation does not map perfectly onto the elements defined by the elaboration likelihood model, though clear correspondences exist. ELM identifies a number of characteristics of a source (i.e. recruiter) that can enhance the likelihood that a receiver (i.e. patient) will form positive attitudes toward the behavior (i.e., prospect of participating in a research study), all of which pertain to the credibility of the source. Credibility is comprised of judgments of a sources' expertise and trustworthiness; expertise is a set of "skills, competencies, and characteristics that enable a party to have influence within some specific domain" [14], while trustworthiness is defined as the ability, benevolence, and integrity of a source that provides health information [14]. ELM-based studies have shown that receivers often make judgments of credibility on the basis of the source's physical appearance or social status [15, 16], so it is not surprising that physical appearance cues like professional clothing, scrubs, or lab coats are often cited as cues that enhance the credibility of recruiters [17–19]. Perceived similarity of personal traits, particularly race (i.e., homophily) also plays an important role in establishing trustworthiness [20]. This principle is reflected in the findings of studies of clinical trial accrual; homophily has been shown consistently to enhance clinical trial recruitment interactions, particularly in the context of minority recruitment [17, 19]. Finally, social science researchers have established that credibility—particularly trustworthiness—is enhanced when receivers believe that the source lacks ulterior motivation to influence the receiver (i.e., absence of commercial motivation) [21]. Although this factor is not broadly represented in the clinical trial communication literature, a study conducted by Schutt et al. [22] indicated that patients are concerned that researchers benefit from patient participation more than patients themselves do [22].

Generally speaking, the literature on clinical trial communication paints a picture of the influence of source credibility using a much broader brush than what would be used by ELM theorists. Further, the elements related to source characteristics that are most frequently cited emerge from separate, unrelated studies. For example, studies frequently cite the importance of homophily between patients and the professionals who recruit them for research participation [5, 23, 24], but these are generally not the same studies that examine the role of perceived expertise of clinical research coordinators [25, 26] or the general preferences of patients for

physicians (the ultimate "experts") to present clinical trial opportunities [27–30]. Similarly, the literature on clinical trial communication clearly emphasizes the importance of "establishing a relationship" with prospective research participants [31–40] but researchers have not linked this recommendation to source credibility. However, establishing a trusting relationship would be difficult to accomplish without the sense that the CRC has the patient's best interest in mind and is not driven by a profit motive. Thus, the way that existing studies on clinical trial recruitment frame the establishment of relationships with participants is consistent with ELM-based definitions of trustworthiness. While few of these studies have used ELM (or really, any theoretical framework), Yang et al., [41] survey-based research indicates that a trusting relationship with doctors predicts systematic processing of information about clinical trial participation.

It is also worth noting that studies on clinical trial communication generally focus either on the perspectives of clinical research coordinators or patients who are recruited into clinical trials. For example, it is not known whether the ways in which patients evaluate the credibility of clinical research coordinators (CRCs) align with the strategies used by CRCs to establish credibility. In this study, we examine the role of expertise and trustworthiness in shaping the attitudes of prospective research participants by analyzing data gathered from both CRCs and Black/African American patients. We seek to address the following research questions:

RQ1: What cues do Black/African American patients use to evaluate the expertise of clinical research coordinators (CRCs) or other recruiters?

RQ2: How do CRCs try to establish expertise with Black/African American patients?

RQ3: What cues do Black/African American patients use to evaluate the trustworthiness of clinical research coordinators or other recruiters?

RQ4: How do CRCs try to establish trustworthiness with Black/African American patients?

## Methods

### Participants

A total of 48 individuals participated in 9 focus groups conducted in a major metropolitan area of the southeastern United States. Patient participants were recruited by the institution's Behavioral and Community-Based Shared Resource (BCSR) while clinical research coordinators were recruited by the authors with the assistance of the institution's Clinical and Translational Science Institute (CTSI). Separate focus groups were conducted with African American patients (*k* = 3), Black Caribbean patients (*k* = 3), and clinical research coordinators (*k* = 3). The demographic composition of the patient and CRC groups appears in Table 1.

### Procedures

Participants were first consented to the study by either one of the study authors (CRC focus groups) or a member of the BCSR staff (patient focus groups). After signing the consent forms, each focus group was asked a series of five questions (see S1 and S2 Files) related to the experiences of the process of either study recruitment (CRCs) or being recruited to research studies and clinical trials (patients). Focus groups were facilitated by a team of two people and included at least one Black Caribbean individual. Focus groups lasted between 50 to 113 minutes. All sessions were audio and/or video recorded to aid the transcription process. Transcriptions of English-language focus groups were performed by several of the manuscript authors; transcriptions of the Haitian Creole focus group were translated by certified medical translators employed by the BCSR. All names used in the manuscript are pseudonyms.

**Table 1. Participant demographics.**

| Demographic Category | | Patient groups (n = 31) | CRC groups (n = 17) |
|---|---|---|---|
| Race | | | |
| | Black or African American | 31 | 3 |
| | White | 0 | 12 |
| | Asian | 0 | 1 |
| | Other | 0 | 1 |
| Ethnicity | | | |
| | Non-Hispanic | 31 | 3 |
| | Hispanic | 0 | 14 |
| Age | | 18–68 ($M = 54.29$) | 23–75 ($M = 43.65$) |
| Gender | | | |
| | Female | 19 | 14 |
| | Male | 11 | 3 |
| | Transgender | 1 | 0 |
| Education | | | |
| | Some high school | 8 | 0 |
| | High school diploma | 9 | 0 |
| | Some college | 5 | 1 |
| | College degree, Associates | 1 | 2 |
| | College degree, Baccalaureate | 3 | 5 |
| | Some post-graduate education | 2 | 2 |
| | Graduate degree, Masters | 2 | 5 |
| | Doctoral degree, PhD/MD/JD | 1 | 2 |

To be eligible for the study, patients had to have had experience being recruited for at least one study prior to the current study, be over 18 years of age, self-identify as African American or Black Caribbean and speak either English or Haitian Creole. Clinical research coordinators had to self-identify as having had significant experience recruiting African American and/or Black Caribbean patients for research studies. All participants received a $40 gift card in exchange for their time.

## Data coding and analysis

All transcripts were uploaded into NVivo (2020 release) for coding and analysis. After multiple readings of all transcripts, all of the study authors discussed the themes they observed and proposed coding categories for the data. The development of the coding scheme was highly iterative and evolved during the early stages of coding. Two of the authors coded all of the data reported on in this manuscript using a modified constant comparative method [42] where the authors first reviewed themes related to characteristics of the person providing study information. The first two authors generated separate codebooks and then again reviewed the data to establish the strongest correspondence between the codes, research questions, and data. Subsequently, both authors coded all of the data; any disagreements in coding were resolved through discussion. Analysis of the data was primarily performed by the first two authors. Preliminary results were presented to the team for discussion and necessary modifications were made.

## Results

### Patients' assessment of expertise (RQ1)

A number of responses from focus group attendees help to address our research question about the cues that African American and Black Caribbean patients use to evaluate the expertise of professionals who approach them with the opportunity to join a clinical trial or research study. These responses are broadly related to impressions of professionalism, including (1)

appearance cues, (2) message attributes, and (3) the recruiter's associated history with the focus of the study. For example, Charles summarized this evaluation simply: "When you're doing like clinical trials or something like that you want somebody serious, professional, [business-like], you know, they're about their business, right. [They let you know] This is my livelihood, right, so that's what I do, right."

While professional manner and personal appearance are influential, it is not the only physical cue that helps to establish expertise, as Louise points out. The credibility of clinical research coordinators' employer is displayed through physical artifacts like recruitment materials and forms as well as the clothing that recruiters wear.

> *Seeing the logo [on the materials], seeing the appearance of the person, seeing the [academic medical institution] logo on the table, it seems more legit. So, it is sometimes appearance, you know, its appearance, has to look good in order to attract me, too. So . . . it's like a combination: look good, how you approach. . . I think that has lots to do with the appearance of the table, the appearance of how and what you're carrying, how you look, do you have a badge on, how legit, all that matters.*

Consistent with assertions in the literature that source factors like expertise interact with message factors to establish credibility, Natalie pointed out that "what is said" is as important as "how it's said" to create judgments about "who's saying it."

> *There are some people that explain things to you. They explain it to you in a way that's good for you so that you end up knowing. But there are other people that come up with some type of authority. About what they know, do you understand? The message does not come through. You are following them because they used a kind of, a way, some type of authority with you. . . .That means take it or leave it. No, medicine does not work that way. [There's an objective truth about] when things are good for the person. There is a [right] way to approach the person to convince them [to join a study].*

In addition to broad perceptions of professionalism, reassurances that the PI or researcher has experience also helps to establish expertise. Alisha, who enrolled in a clinical trial to repair a serious deformity described how the PI's careful study of the problem and past experience reassured her that the PI could be trusted.

> *I was in a clinical trial. . . because I had a really bad jaw defect. And the doctor that I was visiting, he studied in Africa and studying the bone structures of different people. And he had performed the surgery before on multiple people, but it wasn't a common surgery that they do all around. . . . Like, I wasn't skeptical at all, like I trusted him.*

Taken together, these responses indicate that patients want a sense of professionalism in the approach about study participation. While not often done by clinical research coordinators, offering information about the experience and expertise of the study PI may also be helpful to patients.

## Recruiters' strategies to establish expertise (RQ2)

Our second research question asked how research professionals establish expertise with patients (or community members) who they approach about research study participation. These communication strategies can be categorized into (1) displaying physical appearance cues that advance perceptions of credibility and (2) fostering impressions of competence

through the verbal and nonverbal delivery of study information during the approach and consent process.

Clothing and appearance. Clinical research coordinators in our study were highly conscious of the role of clothing in creating positive impressions of professionalism. Will talked about the ways in which medical-related clothing and institutional identification create a greater sense of trust.

> *Since I'm more based in the hospital, I always wear scrubs. So they always respect [me], first of all, because you have to introduce yourself and show your ID and they know that if you're working in the [academic medical center], so they're going to trust you right away. So I think for me, wearing scrubs . . . what you're wearing is advantage.*

Clothing that is branded with the academic medical center's logo also appears to help support impressions of credibility, according to Ozzie. At the same time, being both professional and appropriate for the environment is seen as important.

> *In some studies, they actually have their own shirts that's made up specifically with the study name and everything on it, you know, I've done [that] especially when we're doing outreach, they have different shirts that they've made up to see as [academic medical center]. But it depends now, if I'm going to a meeting in a community, that's different, I'm going to dress like I'm going to work. And at the community, at that meeting is going to be different people from all organization that's going to be there. So that's totally different. But I'm saying when you're actually walking the streets, you know, that's different. You want to blend in, to be very honest.*

Clothing may also signal status to research participants. Interestingly, this sense of status may be transferred to patients who are enrolled in a research study. Maria described this dynamic.

> *When I'm in clinic, I actually I don't wear a lab coat, but I have [a] black jacket and it has our names on it, signs of [academic medical center]. . . . And I, I think it makes them feel better. . . . [O]ur patients, they kind of get fast tracked throughout the entire hospital. And so they're with me or with one of my staff members, and it's kind of like they get that personalized attention. . . . [W]hen it's time for them to do a procedure I personally come to get them, and so kind of they're feeling that VIP status, and then you see other patients like looking at them like —why are they getting [treated special]? They like to feel special, they like to feel special, so I think that I sometimes I just wear that jacket mostly so that they feel that specialness of when I'm coming to get them and everyone's wondering like so is this a VIP patient? . . . I think that helps.*

Clearly, clothing functions as more than a uniform in the context of research study recruitment. Clothing conveys authority and expertise as well as respect for patients and can even confer a certain status through affiliation.

Credibility is established through "flow" of delivery. The other way that clinical research coordinators establish expertise is through the manner of presentation of information about studies. When the presentation of study material goes well, CRCs characterized the interaction as having "flow." This is described in an interaction between two of our participants.

> *Kelly: I think you really have to be present in that communication, and not just be like a robot repeating the same script, you know, you really have to be there with the person and be passionate about what you're talking about. And just, you know, be in tune with that person.*

Will: And of course, you know, the study—you already know how to explain it.

Angela, an experienced clinical research coordinator who manages a team of professionals who recruit for research studies describes the elements of flow, including body language and vocal quality as well as the positive outcomes of an interaction characterized by flow, particularly the receptivity of individuals being recruited. Interestingly, she cautions that flow is more challenging to establish if the recruiter holds biases against the patient or the population to which s/he belongs.

> Angela: Well, like I've supervised a number of coordinators and Research Associates, and I'll watch them like, as I'm training. . . .I'll watch [and] I can see the nervousness. There's not a flow. When you have a flow . . . participants are going to feel your confidence. I can see my [CRCs], their evolution of confidence. And I can see, okay, that's why they didn't get the first initial people. . . .But when there's a flow, . . .there's a receptiveness to the participant that we're targeting. And like, there can be some implicit biases like that can surface. You see it in the body language, like, are they talking to the participant? Like, are you open? Because all of that will show. Are you comfortable even talking to me? Because you can be like, oh, yeah, I am masking it with my voice and my flow of words, but your body language is completely off. So the participant is like, no, because you're probably scared of connecting with me right now. . . . So, your body language is super critical, your voice, your flow is super effective and engaging your audience.

While professional training and mentoring can enhance a recruiter's flow during an interaction, flow may also be a product of a sense of authentic connection with members of the community who are being recruited. At the same time, personal connection and the resulting "flow" in the interaction does not guarantee that a patient will enroll in a study. Sheila echoes a several of Angela's points above.

> You know, I have been disappointed many times when I knew for sure I did a good job. I knew for sure I was going to sign them in. And at the last moment, "I'm not interested." That has happened to me numerous of times, and I feel like "Crap, did I just spend 30 minutes?" You know, I feel like I wasted their time. I wasted my time. But you know, it's just how it goes. . . . You have to be comfortable with what you're telling people. You have to believe in what the study or whatever you're doing. If you're not believing in your stuff, they can see or hear that you're not believing in it. . . .I don't want anybody to think I'm just like, "Oh, I do like my check." (everybody laughs)

Preparation is needed to create flow; when preparation is lacking, CRCs recognize that the interaction is less likely to go well and a recruitment opportunity is likely to have been sacrificed, as illustrated by Maria.

> I have I think 14, 15 coordinators that report to me, and I don't have them consent a patient until they've consented me. And I'm the patient from hell, I will sit there with their consent form. [Offers examples of difficult behaviors]. . .And I think that I've done that simply because, back in the day that has happened to me where questions have popped up that I thought I knew or I should have known, and I wasn't prepared, and it kind of didn't start that relationship well with the patient, because I'm looking like an idiot, because I don't know the protocol, you know, as well as I should. I think that that, that honestly, like, if you're unprepared, it's a disaster from the get go.

Thus, "flow" during the presentation of information about research study presentation is a manifestation of staff members' knowledge and professionalism. Good flow, then, helps to establish the credibility of clinical research coordinators.

## Patients' perceptions of the trustworthiness of clinical research coordinators and other study staff (RQ3)

The participants in our study described how they assessed the trustworthiness of the source of information about study participation. Their assessments were largely based on (1) homophily between patient and study staff member and (2) perceptions of the goodwill of the person recruiting them, including the absence of a commercial motivation.

Homophily between patients and coordinators. Homophily, which refers to a similarity between individuals in an interaction, can take a variety of forms. For some patients, simply sharing a language is fundamental. Natalie, a Haitian Creole speaker, stated succinctly:

*First of all, it's about language. Because when you find someone who speaks the same language as you, you [are] more comfortable with everything they are explaining to you.*

Other participants stated that having staff members reflect the racial composition of the community in which they are recruiting is also important.

*Louise: Who you have come in towards you [is important]. How are you communicating? Because you can't come to, say, I guess, in the Latin community and put an African American in the middle of the Latin community, trying to do surveys . . . if you are going to go in a Latin community, get someone who speaks Spanish. If you're going to put someone in a Black community because you want recruit, put someone who is African American.*

However, homophily isn't everything to patients. Josephine discussed the paramount importance of establishing a good relationship, which can overcome differences in cultural background.

*There has to be a good relationship. When the person appears in front of you, there is a way to welcome the person. . . .They feel good. I can [tell you about a doctor's] visit. I don't speak English and my child [who translates for me] was working somewhere. They referred me to a doctor, a Haitian, so I can be comfortable. [I went to the visit but] I did not feel too comfortable. . . .In the meantime, I have another doctor I used to go to. She is French. Even with another nationality. I felt really well, I felt myself. . . .It's important that you make the person feel comfortable . . .[I feel that] you are not going to hurt me. You are going to treat me well, so I can trust you.*

Perceptions of goodwill. Patients' perceptions of goodwill are linked to a sense that the recruiter has their best interests at heart. Goodwill can be found in interactions that are characterized by kindness and empathy; conversely, our participants told us that some study staff members didn't care about them as human beings. According to Trina:

*They feel like 'cause we get compensated they could talk to us how they wanna talk to us and say what they wanna cause they felt like they [were in charge]. [That] made me feel angry.*

Goodwill is also characterized by a lack of commercial motivation. In other words, when prospective research participants perceive that staff involved with research might financially

benefit from signing them onto a study, participants cannot trust staff to be truthful or to act in their best interest.

> *Joanne: Always, when the person approaches me, the person... never knows the field. ...I'm under the impression that the person, they are benefiting from it. Three quarters of the people, we are under the impression that the person who approaches, it is a benefit, an advantage you are taking from [us]. ...So, I think that it's best that we know. ... They must try to have the [patient] understand that [the recruiter is] not taking advantage.*

This concern was echoed by Robert, who like most other participants, did not distinguish between academic medical center staff and "fly-by-night" recruiters who sign up participants on the street using questionable recruitment practices.

> *So don't get somebody that's just doing it for the money, that you know what I'm saying. Get somebody that, that actually loves what you're doing, love the community, and actually believe in the work that you're doing. You know some of these people are just for money.*

Thus, we see that participants are concerned about whether recruitment staff are motivated by financial self-interest when they make an approach about study participation. Additionally, when patients join studies, they are sometimes mistreated by clinical staff members; this lived experience affects whether they are willing to join future studies. This is no doubt a source of frustration for those who recruit for studies and holds implications for organizational structure and human resource policy within the academic medical research enterprise.

### Ways that clinical research coordinators establish trustworthiness (RQ4)

Clinical research coordinators and other study staff members who recruit participants for clinical trials recognize the importance of establishing their trustworthiness. CRCs articulate a number of communication behaviors that help to create trust. These include (1) communication based on homophily between study staff member and potential participants, (2) expressions of goodwill and personal caring for patients, and (3) truthfulness and transparency in the presentation of study information.

Homophily between coordinators and patients. Participants in our focus groups who recruit patients for clinical trials and research studies believe (as our patient participants do) that homophily is important to the success of the recruitment process. Homophily is described by Angela as a functional feature of the recruitment process where a shared cultural background allows CRCs to be better understood by prospective participants; it also encourages recruiters to be more present in their interactions with patients.

> *If [patients are] looking at someone who looks like themselves, understands them, um has the capabilities to relate exactly well... and is, is—can cater to their type of comprehension, right. It requires the recruiter understanding, "How am I communicating? Am I effectively conveying what the study is about?"*

Cultural competency can also align with language competency. Indeed, being able to present study information in an individual's own language is fundamental to the consent process, as Julie points out.

> *I agree that sometimes the color of your skin or the way that you talk, or some Latinos sometimes feel more comfortable with a Latina person, some African Americans feel more*

*comfortable with African American, some Creole or Haitian people like to hear his language and the way that it's explained they consent . . .[W]e have a person who speak Creole, because sometimes Haitians come and they don't speak any English. So, whenever you try to speak in English, they don't understand the really true [nature] of the study. . . .If the patient doesn't understand the study, you can not commit a patient to get in a study. You need to be sure that the patient understands the risks and the benefit of the study so they can sign. So, if you know that the patient has a lack in the language, you should better find somebody to help you.*

While using medical translators can obviously assist with patients' understanding of the technical details of the study, something is inevitably "lost in translation," as Dayana points out.

*The Hispanic population, they are not used to being part of research. But when you start speaking the same language—in my case in Spanish—they feel like somebody is related to them and they're more open, because most of the [earlier] consultations are with the translator and they do not feel very open or used to it. So, when you approach them in their own language and you give them all the consent [forms] in Spanish they feel more open [to participation].*

Beyond the connection that sharing a common language creates, CRCs report that homophily between recruiter and patient cultivates a sense of trust. Sheila asserts that the sense of familiarity creates a greater degree of comfort among prospective participants which helps to facilitate the interaction.

*For me, it's beneficial for me as a black woman when I'm out in the community recruiting black people. . .It's just that they feel more, more comfortable. My partner is Puerto Rican and um we go out and [recruit] together. . . .They say [to her], "You don't belong here, do you?" But when I go by myself, I don't have that. You feel what I'm saying? In Little Havana or wherever else, [they still] asked me [questions, though] . . . because a lot of black people could be Spanish as well. . .It's just a matter of trusting anybody that looks like us. And then [it happens] right away. Like automatic trust, it's there already, doesn't need to, you don't need to work for that.*

There are a number of explanations for the effect of homophily. For CRCs who are from the same minoritized communities as the patients they recruit, a deep level of understanding of shared oppression allows CRCs to communicate more effectively about the promise that clinical trials and research studies can offer.

*Sheila: I want you to know that Black people are very sensitive toward body language. We had to learn to be, we could see resistance, we can tell when someone is being resistant, just by the body language or shutting down because of body language. Because all but some of us have been through in our life with prejudice and bigotry. . . . So when I'm in a community, I recognize that immediately. So it's my responsibility to put them at ease, to let them know, yes, I have a badge on [but I] also make sure when I go into community, I dress in jeans and stuff to dress down, to be a part of, you know, that's important. And so when I feel that I say, "Listen, I understand how you may feel, and I appreciate that, but listen, let me tell you what the benefit is can be for you" because some of the studies that I've [recruited for] do have some benefits, because if we find something physically wrong, we do contact them and let them know so they can seek help [from] their primary care [physician]. . . .So that's really, I tell them, [I] say you're coming into a place that . . .you know, you can feel comfortable.*

Clearly, homophily is not just about having similar sociodemographic characteristics as members of the population being recruited; CRCs appear to be conscious of the ways in which verbal and nonverbal cues can enhance credibility.

Cultivating perceptions of goodwill. CRCs are challenged to be viewed as trustworthy sources of information about clinical trial participation. One way that they rise to this challenge is through expressions of goodwill. In the face of medical mistrust, clinical research coordinators and other staff members who recruit for research studies look for ways to express goodwill toward research participants as well as the community as a whole.

> *Angela: If you can connect with a participant like, "We come from similar backgrounds and I want to help you." If you look like an outsider and you don't look like one of them, they're going to shut down. And if you tell them, "I'm from the Bronx, this is my community, I want to improve my community and I'm here to help you." And like, "Helping you helps me, we help each other."...Then they're like a little more receptive and they send me other people to recruit from their family or friends.*

Expressions of caring and empathy can also create the sense that the study staff member wants the best for a study participant. Janice has had to work to counter the perception that clinical staff view them as guinea pigs rather than human beings.

> *I can't say how many times in a consenting process that somebody, while they're very much ready, want to do the research, they'll say like an offhand remark. They'll be laughing like, "Oh yeah, . . . I know we're a guinea pig," and I have to be like, "You're not a guinea pig. You are a person. And we care more about you than what is going on with this research. If there's something that happens that's going to happen, that's wrong and we see it coming, we will stop and if we feel like it's best for you to stop."... I have to make it very clear for them to be like the research doesn't take over them as a person as an individual.*

Perceptions of goodwill toward patients and the communities to which they belong are also enhanced or degraded based on whether the recruiter stands to benefit financially from a patient's participation in a study. Juan talked about how patients may have misgivings about his motivations.

> *I think one of the ways [to connect with patients] is like pampering them a little bit. . .they're giving you something, so I guess you have to give back. . .in a sense. I mean. . .they sometimes even ask upfront, like, [whether] I am getting paid, but it's more like that. . . that human level pampering, you know, like, "How you doing? Are you feeling better? How's your day?" [I]t's actually getting down to that level of empathy and getting to know your patient in a personal level.*

Expressions of kindness and empathy and creating a sense of meaningful personal connection are clearly ways that clinical research coordinators establish a sense of goodwill (and thus, trustworthiness) with patients. Because perceptions of goodwill are compromised by the presence of a commercial motivation, it is also important to CRCs to establish with patients that they do not financially benefit from someone's participation, something which might be beneficial to establish when CRCs engage in the recruiting process.

Truthfulness and transparency. CRCs in our study emphasized the importance of truthfulness and transparency as a way to empower patients and build trust. Juan illustrates the importance of using the consenting process emphasize transparency.

*I think I'm just trying to be as transparent, be like, I'm not gonna hide. And I sometimes I'm like, "I am not hiding anything from you. I'm not trying to trick you, I don't want you to feel like you know, you don't have options, you have a lot of options. So if you don't want to do this, you can sign today change your mind, you don't have to sign right now, even after I explained this whole thing. You don't have to sign it". And I think that gives them a little bit . . .[of] the power back that I said that they don't have to do this. They have options, that I'm not lying. I'm not trying to trick them or anything like that.*

Janice is similarly cognizant of the ways that the consenting process can serve to address distrust of the medical system.

*We work in a place, obviously, that is the number one [medical specialty] hospital in the country. We're known for research, yet, we still sometimes kind of get that, you know, distrust of like, "What is it that you're doing to me?" . . .When we go through the consent form, like I said, it's always really about giving the patient the power. I think that when they feel powerful, they'll stick through things. . . .When we're going through a consent, I'll let them know, you know, I'm your primary person of contact, but if you feel like I'm not addressing your concerns, there's this phone number on page 18 that you can go and you can file a complaint. And I think that that lets them know, it's like, "Okay, I'm not just stuck here, like I have a means of like getting out of doing something."*

Managing expectations is another way that study staff members create greater trust with research participants. Although official consent forms disclose the processes involved, the experience is not often realistically presented, as Maria indicates:

*I think explaining [every part of]. . . every procedure before it starts [helps patients]. . . .So, the fact that they go into a specific room, and they know what's going to happen beforehand, makes them more comfortable. When they get out, I'm there, I'm like, "You see, how's everything?" [And they say] "You know, everything was all right." And like, there were no surprises. . . Even though we go through the [technical information on the] informed consent form, it's different when it's happening in 30 seconds. Like, "Okay, we're going to start this, and this is [what is] going to happen." I think that helps also in terms of making that connection with that patient.*

Trustworthiness, as a fundamental core component of credibility, is cultivated by clinical research coordinators through three primary communication strategies. First, CRCs try to recruit within communities that they are members of, which provides the cultural sensitivity needed to be effective and appropriate in their approach. Second, CRCs attempt to establish goodwill with their patients through expressions of empathy and support and by clarifying that they do not financially benefit from successful recruitment of participants. Third, CRCs report that they engage with patients in ways that are characterized by transparency and truthfulness; in turn, this creates trust and a sense of empowerment, which is a powerful antidote to medical mistrust.

## Discussion

The goal of this study was to explore the role of source credibility in recruitment of African American and Black Caribbean patients to research studies and clinical trials. The elaboration likelihood model (ELM) proved to be a useful organizing framework for the perspectives of both patients and CRCs. Because ELM is a model that describes how attitudes are formed and

changed, we can see the ways that source characteristics like expertise and perceived trustworthiness can motivate patients to process complicated information about study participation. Consent forms can be 20 (single-spaced!) pages or more, containing notoriously difficult-to-understand language. Clinical research coordinators are often quite talented at providing oral "translations" and explanations of official study information, but unless participants are driven by a desire for financial incentives or significant personal benefits, patients' perceptions of the credibility of CRCs are likely driving much of the success of recruitment efforts.

Our research questions offer the opportunity to compare and contrast the responses of patients with those of clinical research coordinators. Is credibility established in the minds of patients in the same ways that clinical research coordinators believe that it is? It appears that this is generally the case. While CRCs spend a great deal of time thinking about how to present study information effectively, patients have few reasons to consider these questions, so it is not surprising that the CRCs in our study provide more detailed information related to credibility. In this study, we discovered that clinical research coordinators and African American and Black Caribbean patients had very similar (though not identical) views of the importance of cues that support evaluations of the credibility of the source of information. Patients' perceptions of expertise were drawn from global assessments of the professionalism of staff members. CRCs, on the other hand, articulated more specific strategies for cultivating perceptions of expertise. These included physical cues like professional (or otherwise context-appropriate) clothing and branding of recruitment materials (e.g. institutional logo on consent forms). Expertise was also established through the "flow" of delivery of information, which could be accomplished only after thorough intense preparation. When CRCs were deeply familiar with all study materials and processes and rehearsed their approach, including responses to common questions, they were more likely to achieve flow in their interactions.

Trustworthiness, the other core element of source credibility, was viewed similarly by both CRCs and patients. Homophily (i.e. cultural, racial, and/or linguistic similarity) and expressions of goodwill were cited by members of both groups as being linked to the development of a sense of trust. Both patients and CRCs indicated that trust comes more easily when CRCs and patients share a deep understanding of one another; this trust is enhanced when CRCs demonstrate empathy for patients' circumstances and can provide reassurances that there are no ulterior financial motives driving the study recruitment process. Additionally, CRCs discussed the importance of transparency in supporting perceptions of trustworthiness. CRCs took great pains to accurately depict the study experience, for example, as a way to build trust, which they believed enhanced study retention and future recruitment.

There are several limitations of our study that are important for future researchers to consider. The inclusionary criteria for our study, particularly for patients, were quite broad. Patients who had had the experience of being recruited for one or more studies in the past were eligible to participate in our study. Our intention was to include voices from individuals who had refused to participate in research in the past rather than only those who were enthusiastic about the benefits of research or cheerleaders for a specific study or physician. However, we would have benefitted from the insights of patients with a broader array of positive and negative study recruitment experiences. Second, while this is the first study of clinical trial communication that includes a significant number of Black Caribbean participants, it was clear that experiences vary considerably based on socioeconomic status, education level, and English language proficiency. Based on this, it would be impossible to generalize to the broader (and highly diverse) Black Caribbean community or make recommendations about clinical trial communication that are specific to members of this population. Third, as with much qualitative research, we would have benefitted from a larger number of participants, particularly patients. Recruitment challenges and expenses associated with research staff members' time

could have been mitigated by the availability of significant external funding. Fourth, our participant groups—CRCs and patients (and other community-based participants)—had very different demographic compositions. The CRCs who participated in our study were highly educated and largely Hispanic while our other groups were entirely African American and Black Caribbean and included individuals with less education. While this is reflective of both our deliberative sampling strategies (in the case of patient groups) and the composition of CRCs at this institution, differences in both race/ethnicity and other factors like education level and even gender can be expected to have shaped participants' responses to our questions.

In addition to the development and evaluation of CRC training programs focused on communication skills, future research should consider the ways that community outreach efforts could be enhanced by patient education. While the number of African Americans being approached to participate in clinical trials has never been higher (Occa, 2022), it is also clear that increased recruitment, likely part of a broader effort to oversample to try to achieve greater representation, is not necessarily translating into study enrollment or retention. Part of the issue may be due to difficulties determining whether study participation is safe or beneficial. Helping community members distinguish between high- and low-credibility sources of information about research study participation would help them provide meaningful consent (or refusal) to future studies. Our results indicated that a number of patients do not know the difference between academic medical researcher and for-profit research operations that use untrained individuals to recruit participants on the street.

## Ethical considerations

Bioethicists may want to consider the reality of how patients, particularly those with lower levels of education, make decisions about clinical trial and research study participation. While there is increasing attention paid to the language used in informed consent forms, efforts to reduce the reading level of forms does not change the fact that many patients rely on the advice of experts (real or perceived) to make health-related decisions because they lack the ability to make a truly informed decision about what is best for them based on the information provided. By necessity, patients look to community members, family and friends, their physicians, and clinical staff members for information that will help them make decisions. Helping patients assess the credibility of the source of information may help improve the quality of decisions about research study participation.

Physicians are often the first (and most credible) source of information about treatment trials, with CRCs following up to fully consent a patient. Their determination of the suitability of a trial for a patient is a manifestation of a physician's expertise. However, while the expertise of the source is a powerful persuasive force, it is incumbent on CRCs to balance the recommendation of a physician by fully and clearly presenting all study information. Our study does not examine the potential tension between the recruitment process and the consenting process when physicians present a study opportunity to patients, but future research may want to examine this dynamic more closely.

*Additionally, while it is important to "translate" complex scientific terminology into easy-to-understand language and to present the benefits of participation along with the risks, it is imperative that CRCs and other medical professionals who present information about research studies maintain equipoise. Prospective participants must emerge from a discussion with the understanding that there is no preferred assignment to a study condition; research is done precisely to determine whether there is an advantage to a specific treatment or protocol. Further, the fact that our study identified verbal and nonverbal cues that motivate*

*information processing should not be taken guidance on how to manipulate prospective participants to enroll in a study or clinical trial.*

*Finally, there are also concerns that can be raised about the recommendations that emerged from our research regarding professionalism. Definitions of professional attire, for example, need to take care not to disadvantage individuals from less affluent backgrounds. Offering free uniforms (whether polo shirts with the logo of an academic medical center or specially-designed lab coats), medical centers can help CRCs project professionalism without imposing standards that are classist or sexist or which might disadvantage members of the LBGTQIA community.*

## Conclusion

The paramount importance of source credibility in the processes of attitude formation and persuasion has been well-established, dating back at least to Aristotle's *Rhetoric*. The goal of our current study is to explore the role of perceived credibility on patients' willingness to consider the possibility of clinical trial participation. Because careful consideration of information about research participation is central to the informed consent process, the role of source credibility in "nudging" individuals toward greater elaboration should be better understood. While there is no way for us to know which cognitive route patients used when CRCs presented information or whether patients ultimately elaborated on this information, our data indicate that a range of source characteristics are likely to lead to elaboration. This specific information will be vital in the development of future professional training programs to enhance the effectiveness of CRC communication behaviors. Such training programs have long been called for, but few have been developed in reference to the empirical literature and even fewer have been rigorously evaluated. By including diverse CRCs as well as African American and Black Caribbean patients in our study, the development of future programs can be based on a wider range of perspectives and insights and experiences which should enhance the potential impact of those programs, particularly in minority and underserved communities.

## Supporting information

**S1 File. Focus group questions for patients.**
(DOCX)

**S2 File. Focus group questions for clinical research coordinators.**
(DOCX)

## Author Contributions

**Conceptualization:** Susan E. Morgan, Tyler R. Harrison, Kallia O. Wright, Xiaofeng Jia, Bonnie Deal, Kate Malova.

**Formal analysis:** Susan E. Morgan, Tyler R. Harrison, Kallia O. Wright, Xiaofeng Jia, Bonnie Deal, Kate Malova.

**Funding acquisition:** Susan E. Morgan, Tyler R. Harrison.

**Investigation:** Susan E. Morgan, Tyler R. Harrison, Kallia O. Wright, Xiaofeng Jia, Bonnie Deal, Kate Malova.

**Methodology:** Susan E. Morgan, Tyler R. Harrison, Kallia O. Wright.

**Project administration:** Susan E. Morgan, Tyler R. Harrison, Kallia O. Wright.

**Resources:** Susan E. Morgan, Tyler R. Harrison.

**Supervision:** Susan E. Morgan, Tyler R. Harrison, Kallia O. Wright.

**Validation:** Tyler R. Harrison.

**Writing – original draft:** Susan E. Morgan, Tyler R. Harrison.

**Writing – review & editing:** Susan E. Morgan, Tyler R. Harrison, Kallia O. Wright, Xiaofeng Jia, Bonnie Deal, Kate Malova.

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
