## [Decision Letter · Decision Letter 0]

14 Feb 2023

PONE-D-22-26524The role of perceived expertise and trustworthiness in research study and clinical trial recruitment: Perspectives of clinical research coordinators and African American and Black Caribbean patientsPLOS ONE

Dear Dr. Morgan,

Thank you for submitting your manuscript to PLOS ONE. After careful consideration, we feel that it has merit but does not fully meet PLOS ONE’s publication criteria as it currently stands. Therefore, we invite you to submit a revised version of the manuscript that addresses the points raised during the review process.

We look forward to receiving your revised manuscript.

Kind regards,

Federica Canzan

Academic Editor

PLOS ONE

Journal Requirements:

“This study received funding support from the University of Miami Clinical Translational Science Institute, UL1 TR000460 (SEM), http://miamictsi.org, and the University of Miami’s Provost’s Research Award, PRA2022-2510 (SEM, TRH), https://www.research.miami.edu/about/admin-areas/rde/provosts-awards/provosts-research-awards/index.html. The funders had no role in study design, data collection and analysis, decision to publish, or preparation of the manuscript.”

4**. **Please include your full ethics statement in the ‘Methods’ section of your manuscript file. In your statement, please include the full name of the IRB or ethics committee who approved or waived your study, as well as whether or not you obtained informed written or verbal consent. If consent was waived for your study, please include this information in your statement as well.

5 Please review your reference list to ensure that it is complete and correct. If you have cited papers that have been retracted, please include the rationale for doing so in the manuscript text, or remove these references and replace them with relevant current references. Any changes to the reference list should be mentioned in the rebuttal letter that accompanies your revised manuscript. If you need to cite a retracted article, indicate the article’s retracted status in the References list and also include a citation and full reference for the retraction notice.

Reviewers' comments:

Reviewer's Responses to Questions

**Comments to the Author**

1. Is the manuscript technically sound, and do the data support the conclusions?

Reviewer #1: Yes

Reviewer #2: Yes

2. Has the statistical analysis been performed appropriately and rigorously? 

Reviewer #1: N/A

Reviewer #2: N/A

3. Have the authors made all data underlying the findings in their manuscript fully available?

Reviewer #1: Yes

Reviewer #2: Yes

4. Is the manuscript presented in an intelligible fashion and written in standard English?

Reviewer #1: Yes

Reviewer #2: Yes

5. Review Comments to the Author

Reviewer #1: Thank you for the opportunity to review this paper. It’s an interesting study with lots of scope for the findings to positively influence the way that trials recruit potential participants. I will be glad to see this published, with a few tweaks to make it as useful as possible for readers. Please see itemised comments below.

1. In the Introduction you state that African Americans are approached more frequently to join trials - this is a surprise to me, in the UK I don’t think this is the case. Could you add in a sentence or two about why this is, or at least an additional reference to support this figure. In the Discussion of the paper you reference, the authors comment on being surprised by the figure of 16% with some indications about why this is.

2. In the Introduction you state that ‘clinical trials offer access to cutting-edge care and treatments that cannot be accessed in any other way’ - this isn’t necessarily true of pragmatic trials, academic trials, and later phase trials, and the sentence should reinforce that this is usually the case for earlier phase trials.

3. I’m not familiar with the elaboration likelihood model but the section (line 68 onwards) where you describe it is well written and clear, providing both pros and cons of the model as well as reasons why you have chosen to use it. Could you also add in why you chose this model over others that may have also been effective in this study?

4. The participant demographics are clearly split between majority Black or African American and Non-Hispanic for patient groups, and majority white and Hispanic for CRC groups, gender and education is also split different in the two participant groups - this need a comment in the Discussion.

5. Line 212 - suggest removing ‘emerging themes’, themes do not emerge, themes are constructed and shaped by the interpretation of the researcher(s) doing the analysis.

6. Please make sure there are quotation marks around all quotes in the Results section, there are some missing, particularly in the bigger quotes e.g., line 234-239, line 244-251.

7. The comments on professionalism, clothing and appearance etc in terms of patients’ assessment of expertise are quite narrow in their definition of professionalism - ‘business-like’ and comments on personal appearance linked to authority and expertise. This has the potential to veer into misogyny and other forms of discrimination. This deserves a comment in the Discussion about how we define ‘professionalism’ in modern society, how that might impact women for example, younger PIs, neurodiverse PIs, people from LGBTQIA+ communities, and others.

8. I’d be keen to see a section on ethics and equipoise added to the Discussion of the manuscript. The perceptions detailed throughout the Results section highlight the importance of maintaining equipoise throughout the recruitment process and its interactions. Without highlighting this there’s a risk that people go on to implement strategies to increase trust, cultivate perceptions of goodwill etc and forget the fundamentals of equipoise - the fact that we are doing a trial means that we do not know whether the trial’s intervention will be effective or not, and we must make that clear to potential participants. The non-verbal queues discussed in this paper have the potential to negate that equipoise.

Reviewer #2: This is a well written paper, easy to follow and on a very important and timely topic. It definitely adds to the literature on the source of trial information and its role in trustworthiness. I would have ideally liked to see the role of physicians in the context of trustworthiness because for some diseases (cancer) physicians are gatekeepers- they introduce the trial and then have the coordinators go through the consent and provide further elaboration. Consider adding this to the discussion.

6. PLOS authors have the option to publish the peer review history of their article (what does this mean?). If published, this will include your full peer review and any attached files.

Reviewer #1: **Yes: **Dr Heidi Gardner

Reviewer #2: **Yes: **Soumya Niranjan

---

## [Author Response · Author response to Decision Letter 0]

17 Mar 2023

Authors’ Response to Reviewer #1: 

Thank you for the opportunity to respond to the points that you made in your careful review of the paper. We believe that we have addressed each point. 

1. In the Introduction you state that African Americans are approached more frequently to join trials - this is a surprise to me, in the UK I don’t think this is the case. Could you add in a sentence or two about why this is, or at least an additional reference to support this figure. In the Discussion of the paper you reference, the authors comment on being surprised by the figure of 16% with some indications about why this is.

Authors’ Response: We are encouraged by the new data that shows that African Americans are being approached more frequently to join clinical trials. The new data (published just last year) comes from the large-scale, nationally representative HINTS dataset. We offer a bit of speculation about the reason for this in the discussion section, as you have suggested. We believe that this increase is due to deliberate oversampling of African Americans in an attempt to ensure that the final group of study participants is more closely representative of the U.S. population.

2. In the Introduction you state that ‘clinical trials offer access to cutting-edge care and treatments that cannot be accessed in any other way’ - this isn’t necessarily true of pragmatic trials, academic trials, and later phase trials, and the sentence should reinforce that this is usually the case for earlier phase trials.

Authors’ Response: Thank you for the suggestion to clarify this for our readers. We have added this clause to the sentence where it appears.

3. I’m not familiar with the elaboration likelihood model but the section (line 68 onwards) where you describe it is well written and clear, providing both pros and cons of the model as well as reasons why you have chosen to use it. Could you also add in why you chose this model over others that may have also been effective in this study?

Authors’ Response: Based on your suggestion, we have added information in the introduction leading up to the review of the elaboration likelihood model that alerts readers to another possible theoretical framework that is also widely respected: the heuristic systematic model (HSM). Both frameworks were developed in the same time period (late 1970s/early 1980s) and both have a considerable amount of empirical evidence in support of their principles. The similarities between the two frameworks far outweigh the differences. As best as I can determine, ELM seems to be preferred by health communication researchers while HSM seems to be preferred by social psychologists, perhaps because of the greater emphasis on the measurement of prior attitudes and in-depth assessment of motivations to process information. Because of the broad nature of qualitative research and our lack of quantitative measurement of prior attitudes (and motivations), we decided that ELM is the better fit.

4. The participant demographics are clearly split between majority Black or African American and Non-Hispanic for patient groups, and majority white and Hispanic for CRC groups, gender and education is also split different in the two participant groups - this needs a comment in the Discussion.

Authors’ Response: We agree that these differences in demographic factors have surely shaped their responses to the questions we posed in our study. We have added several sentences to our discussion section on the importance of recognizing these differences. We appreciate the suggestion.

5. Line 212 - suggest removing ‘emerging themes’, themes do not emerge, themes are constructed and shaped by the interpretation of the researcher(s) doing the analysis.

Authors’ Response: As suggested, we have removed the term “emerging” from the Data Coding and Analysis section.

6. Please make sure there are quotation marks around all quotes in the Results section, there are some missing, particularly in the bigger quotes e.g., line 234-239, line 244-251.

Authors’ Response: We have gone back through the paper to ensure that all quotation marks appear where they should. Per APA conventions, we use quotation marks only for in-text quotations, while block quotes set apart from the body of the text do not require quotation marks.

7. The comments on professionalism, clothing and appearance etc in terms of patients’ assessment of expertise are quite narrow in their definition of professionalism - ‘business-like’ and comments on personal appearance linked to authority and expertise. This has the potential to veer into misogyny and other forms of discrimination. This deserves a comment in the Discussion about how we define ‘professionalism’ in modern society, how that might impact women for example, younger PIs, neurodiverse PIs, people from LGBTQIA+ communities, and others.

Authors’ Response: We agree that the potential for sexist, classist, and anti-queer discriminatory impact exists. We have added cautionary statements to this effect in the Discussion section, as suggested.

 8. I’d be keen to see a section on ethics and equipoise added to the Discussion of the manuscript. The perceptions detailed throughout the Results section highlight the importance of maintaining equipoise throughout the recruitment process and its interactions. Without highlighting this there’s a risk that people go on to implement strategies to increase trust, cultivate perceptions of goodwill etc and forget the fundamentals of equipoise - the fact that we are doing a trial means that we do not know whether the trial’s intervention will be effective or not, and we must make that clear to potential participants. The non-verbal queues discussed in this paper have the potential to negate that equipoise.

Authors’ Response: This is an excellent suggestion and we have responded by creating a section dedicated to bioethical concerns in the Discussion section.

 

Authors’ Response to Reviewer #2: 

Thank you for reviewing our paper and for your favorable review. We have responded to your critique as detailed below. 

This is a well written paper, easy to follow and on a very important and timely topic. It definitely adds to the literature on the source of trial information and its role in trustworthiness. I would have ideally liked to see 

the role of physicians in the context of trustworthiness because for some diseases (cancer) physicians are gatekeepers- they introduce the trial and then have the coordinators go through the consent and provide further elaboration. Consider adding this to the discussion.

Authors’ Response: We appreciate this excellent suggestion. We have added this point to a new section on bioethics that is part of the Discussion section.

---

## [Editor Report · Decision Letter 1]

12 Apr 2023

The role of perceived expertise and trustworthiness in research study and clinical trial recruitment: Perspectives of clinical research coordinators and African American and Black Caribbean patients

PONE-D-22-26524R1

Dear Dr. Morgan,

We’re pleased to inform you that your manuscript has been judged scientifically suitable for publication and will be formally accepted for publication once it meets all outstanding technical requirements.

Kind regards,

Federica Canzan

Academic Editor

PLOS ONE
---

## [Editor Report · Acceptance letter]

26 Apr 2023

PONE-D-22-26524R1 

The role of perceived expertise and trustworthiness in research study and clinical trial recruitment: Perspectives of clinical research coordinators and African American and Black Caribbean patients 

Dear Dr. Morgan:

I'm pleased to inform you that your manuscript has been deemed suitable for publication in PLOS ONE. Congratulations! Your manuscript is now with our production department. 

Kind regards, 

on behalf of

Professor Federica Canzan 

Academic Editor

PLOS ONE